# A Thermal-Fluid-Solid Coupling Computation Model of Initiation Pressure Using Supercritical Carbon Dioxide Fracturing

**Yi Chen** [1,2,3] 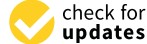, **Zhihong Kang** [1], **Yuzhu Kang** [4], **Xiaocheng Chen** [5], **Xiaohong Chen** [6,7], **Qingteng Fan** [8], **Yukun Du** [6,7,*] **and Jinguang Wang** [9]

1   School of Energy Resources, China University of Geosciences (Beijing), Beijing 100083, China
2   Oil & Gas Survey, China Geological Survey, Beijing 100029, China
3   Key Laboratory of Unconventional Oil & Gas, China Geological Survey, Beijing 100029, China
4   Petroleum Exploration and Production Research Institute, SINOPEC, Beijing 100083, China
5   Qingdao Geological Exploration Institute, China Metallurgical Geology Bureau, Qingdao 266000, China
6   Key Laboratory of Unconventional Oil & Gas Development, Ministry of Education, Qingdao 266580, China
7   School of Petroleum Engineering, China University of Petroleum, Qingdao 266580, China
8   CRRC Qingdao Sifang Rolling Stock Research Institute Co., Ltd., Qingdao 266000, China
9   No.4 Drilling Company of Daqing Drilling Engineering Company, Daqing 163000, China
*   Correspondence: duyukun@upc.edu.cn

**Abstract:** With the characteristics of low fracturing pressure, little damage to the reservoirs, and assuming the role of carbon storage, supercritical carbon dioxide (SC-CO$_2$) fracturing is suitable for the development of unconventional oil and gas resources. Based on the tensile failure mechanism of rocks, this paper establishes a thermal-fluid-solid coupling initiation pressure model for SC-CO$_2$ fracturing. Using this model, the changes in formation temperature and pore pressure near a wellbore caused by invasion of CO$_2$ into the formation are analyzed, as well as the impact of these changes on the tangential stress of reservoir rocks. The field data of SC-CO$_2$ fracturing in a sandstone gas well are used to validate the reliability of the model. The results show that SC-CO$_2$ fracturing can significantly reduce the initiation pressure, which decreases with the increase in fracturing fluid injection rate. The minimum value of tangential stress is located at the well wall, and the direction of tangential stress caused by formation temperature and pore pressure is opposite, with the former greater than the latter. The increase in Poisson's ratio, the increase in elastic modulus and the decrease in bottom hole temperature can reduce the initial fracturing pressure of the reservoir. The computation model established in this paper provides an effective method for understanding the reservoir fracturing mechanism under the condition of SC-CO$_2$ invasion.

**Keywords:** SC-CO$_2$; hydraulic fracturing; initiation pressure; tangential stress; thermal-fluid-solid coupling

## 1. Introduction

Unconventional oil and gas resources in China are characterized by tight reservoir, complex geological structure, and high clay mineral content, which increases the difficulty of unconventional oil and gas field development [1–4]. Practice shows that large-scale hydraulic fracturing is one of the most effective technical means in unconventional oil and gas development fields [5,6]. However, the traditional hydraulic fracturing poses problems such as high water consumption, reservoir damage, and flowback fluid pollution [7], which restrict the scale and effectiveness of reservoir fracturing, and hinder the process of commercial development of unconventional oil and gas resources in China. Therefore, it is urgent to explore a new environmentally friendly water free fracturing technology.

SC-CO$_2$ fracturing is a new technology that uses SC-CO$_2$ (critical point: temperature 304.1 K, pressure 7.38 MPa) as the fracturing fluid to fracture the reservoir [8,9]. During

fracturing, liquid $CO_2$ is injected into the wellbore at a high pressure and low temperature and heated by the formation around the wellbore when flowing down the wellbore. When it reaches the bottom of the wellbore, it is usually in a supercritical state [10]. In this case, $CO_2$ assumes the characteristics of high density, low viscosity, strong permeability, etc. Compared with water-based fracturing fluid, it is easier to induce secondary fractures and form a complex fracture network [11,12], which makes promising application prospects in fracturing and stimulation of unconventional oil and gas reservoirs. In addition, SC-$CO_2$ fracturing has advantages in reducing crude oil viscosity, competing with methane in terms of adsorption, clay expansion inhibition, water resources conservation, and effective burial of $CO_2$ [13]. Therefore, it is a green and efficient fracturing technology.

The mechanism of reservoir initiation is the inception and primary point of the basic theoretical research of SC-$CO_2$ fracturing which can be traced back to the 1950s. At present, it is principally based on Hubbert–Willis model and Haimson–Fairhurst model. Hubbert et al. [14] proposed a near-wellbore stress model (Hubbert–Willis model) affected by wellbore pressure based on triaxial compression experiments in 1957 and provided criteria for wellbore fracture mode, but did not take into consideration the influence of radial seepage. In 1967, Haimson et al. [15,16] introduced the seepage term into the wellbore stress model and deduced the calculation formula of the starting pressure and fracture width. Huang [17] further proposed the criteria for the initiation and propagation of vertical and horizontal fractures, and concluded that lithology, interface, and fracturing fluid can affect the initiation and propagation of fractures. In 2000, Hossain et al. established the initiation criterion of hydraulic fracturing for perforated wells by using coordinate transformation and superposition principle [18]. Subsequently, scholars gradually carried out research on the effects of the well deviation and orientation [19], completion and perforation [20,21], formation structure and bedding structure [22,23], and reservoir anisotropy and natural fracture distribution [24,25] on the initiation pressure and orientation. However, when SC-$CO_2$ is used as a fracturing fluid, its high permeability and interaction with rock significantly affects the stress state around the well, thereby changing the fracturing pressure. Laboratory experiments show that SC-$CO_2$ fracturing has lower initiation pressure and more complex fracture network systems than conventional hydraulic fracturing [26,27]. Chen et al. [28] established a prediction model for the initiation pressure of SC-$CO_2$ fracturing considering such factors as SC-$CO_2$ viscosity, fluid compressibility, and pressurization rate, and conducted parameter sensitivity analysis. Xiao et al. [29] coupled wellbore flow equation, wellbore pressurization rate, and initiation pressure equation, proposed a $CO_2$ fracturing initiation pressure model considering real bottom hole temperature and pressure conditions, and proposed the optimal injection rate range. Zhong et al. [30] established a thermal-fluid-solid coupling model considering wellbore temperature pressure field and stress field around the well and analyzed the change rule of SC-$CO_2$ fracturing time with factors such as initial injection rate, injection pressure, and injection temperature.

In summary, the research on the fracturing mechanism of SC-$CO_2$ fracturing is predominantly confined to laboratory experiments, and a theoretical system of reservoir fracturing considering the impact of $CO_2$ invasion has yet to be formed. In view of the shortcomings of the existing theoretical models, this paper attempts to establish a thermal-fluid-solid coupling SC-$CO_2$ fracturing initiation pressure model, taking into full account of the impact of SC-$CO_2$ on the near-wellbore formation temperature, pore pressure, and rock tangential stress. By analyzing the stress weakness of the wellbore and near-wellbore formation, the authors aim to calculate the initiation pressure and provide theoretical guidance for on-site construction.



## 2. Theoretical Model

### 2.1. Tangential Stress Model around Well

The fracture is related to tensile failure during fracturing. Taking the compressive stress as positive, the condition for formation fracturing is as follows:

$$\sigma_\theta \leq -\sigma_t \tag{1}$$

where $\sigma_\theta$ is effective tangential stress, MPa; $\sigma_t$ is the tensile strength of reservoir rock, MPa.

$CO_2$ can invade the formation after reaching the bottom of the well for its strong permeability, and then change the pore pressure around the well and affect the physical properties of the rock, thus change the tangential stress around the well. At the same time, the temperature difference between the fluid and the reservoir also causes the change in the tangential stress around the well.

According to the analysis above, the tangential stress around the well during SC-$CO_2$ fracturing is mainly caused by four basic loads: external radial stress, wellbore pressure, pore pressure, and thermal effect. According to the superposition principle [24], the expression of tangential stress around the well is:

$$\sigma_\theta = \sigma_{\theta 1} + \sigma_{\theta 2} + \sigma_{\theta 3} + \sigma_{\theta 4} \tag{2}$$

where

$$\begin{cases} \sigma_{\theta 1} = \frac{\sigma_H + \sigma_h}{2}\left(1 + \frac{R_w^2}{r^2}\right) - \frac{\sigma_H - \sigma_h}{2}\left(1 + 3\frac{R_w^4}{r^4}\right)\cos 2\theta \\ \sigma_{\theta 2} = -p_w \frac{R_w^2}{r^2} \end{cases} \tag{3}$$

where $\sigma_{\theta 1}$ is the tangential stress caused by external radial stress, MPa; $\sigma_{\theta 2}$ is the tangential stress caused by the wellbore pressure, MPa; $\sigma_{\theta 3}$ is the tangential stress caused by fluid seepage effect in the wellbore, MPa; $\sigma_{\theta 4}$ is the tangential stress caused by thermal effect, MPa; $\sigma_H$, $\sigma_h$ is the maximum and minimum horizontal principal stress, respectively, MPa; $R_w$ is the borehole radius, m; $r$ is the distance from the borehole in the horizontal direction, m; $p_w$ is the bottom hole pressure, MPa; and $\theta$ is the well circumference angle, (°).

The mud cake cannot be formed inside the well wall for $CO_2$ has little wall building property [31]. At the same time, the bottom hole pressure increases during injection, and the pore pressure increases caused by invasion of SC-$CO_2$ into the formation, thus the tangential stress changes. The pore pressure change $\Delta p_p$ is defined as

$$\Delta p_p(r, t) = p_p(r, t) - p_{po} \tag{4}$$

where $p_p(r,t)$ is the formation pore pressure around the well, MPa; $p_{po}$ is the original formation pressure, MPa; and $t$ is the time, s.

The tangential stress distribution formula caused by the change in pore pressure is

$$\begin{cases} \sigma_{\theta 3} = -\frac{2\eta}{r^2}\int_{R_w}^{r} r' \Delta p_p(r', t)dr' + 2\eta \Delta p_p(r, t) \\ \eta = \frac{G}{\lambda + 2G}\alpha = \frac{1 - 2\nu}{2(1 - \nu)}\alpha \end{cases} \tag{5}$$

where $\eta$ is the stress coefficient of porous elastic medium; $G$ and $\gamma$ are the Lame coefficient of rock; $\nu$ is Poisson's ratio; $\alpha$ is the Biot coefficient; and $r'$ is the variable in the integral.

The thermal stress is generated when there is a temperature difference between the fluid and the reservoir, which causes rock damage [32,33]. During SC-$CO_2$ fracturing, there is a temperature difference between the injected low temperature $CO_2$ and the reservoir when it reaches the bottom of the well, and the resulting thermal stress affects the tangential stress around the well. Define the temperature change field $\Delta T$ of rock around the well as

$$\Delta T(r, t) = T(r, t) - T_0 \tag{6}$$

where $T(r,t)$ is the formation temperature around the well, K; and $T_0$ is the original formation temperature, K.

Then the tangential stress caused by temperature difference in the wellbore is

$$\sigma_{\theta 4} = -\frac{E}{3(1-v)}\alpha_m\left[\frac{1}{r^2}\int_{R_w}^r \Delta T(r',t)r'dr' - \Delta T(r,t)\right] \tag{7}$$

where $E$ is the elastic modulus of reservoir rock, MPa; and $\alpha_m$ is the volume thermal expansion coefficient of rock, $K^{-1}$.

If $r = R_w$, the effective tangential stress at the wellbore under the influence of multiple factors is

$$\sigma_\theta = \sigma_H + \sigma_h - 2(\sigma_H - \sigma_h)\cos 2\theta - p_w - 2\eta(p_{po} - p_w) + \frac{E}{3(1-v)}\alpha_m(T_w - T_0) \tag{8}$$

where $Tw$ is the temperature when the fluid reaches the bottom hole, K.

Fracturing occurs when the effective tangential stress in the borehole wall reaches the tensile strength of the reservoir rock. Therefore, it can be calculated that the fracture initiation pressure pf is

$$p_f = p_w = \frac{3\sigma_h - \sigma_H - 2\eta p_{po} + \frac{E}{3(1-v)}\alpha_m(T_w - T_0) + \sigma_t}{2 - 2\eta} \tag{9}$$

### 2.2. Reservoir Temperature and Pore Pressure Model

It can be seen from Equations (5) and (7) that the tangential stress of the near-wellbore formation is related to formation temperature and pore pressure. Therefore, a reservoir temperature and pressure model with mechanical and thermal coupling is established to analyze the initiation conditions of fractures after $CO_2$ invasion.

Based on the mass conservation equation and energy conservation equation of fluid in pores, it can be concluded:

$$\begin{cases} \frac{\partial(\phi\rho_f)}{\partial t} + \nabla \cdot (\rho_f J_v) = 0 \\ \rho_s c_{ps}\frac{\partial T}{\partial t} - \nabla \cdot (k_T\nabla T) - Q_s = 0 \end{cases} \tag{10}$$

where

$$J_v = -\frac{k}{\mu}\nabla p + K_T\nabla T \tag{11}$$

where $\rho_f$ is the density of carbon dioxide, $kg/m^3$; $\phi$ is porosity; $\rho_s$ is the density of reservoir rock, $kg/m^3$; $c_{ps}$ is the specific heat capacity of reservoir rock, J/(kg·K); $k_T$ is the thermal conductivity coefficient, W/(m·K); $Q_s$ is the heat source intensity, J/($m^3$·s); and $J_v$ is the fluid velocity, m/s.

After $CO_2$ invades the reservoir, the fluid velocity is affected by temperature and pressure. The velocity divergence is

$$\nabla J_v = -\frac{k}{\mu}\nabla^2 p + K_T\nabla^2 T \tag{12}$$

where $K$ is permeability, $\mu m^2$; $\mu$ is viscosity, $\mu Pa\cdot s$; $p$ is pore pressure, MPa; and $K_T$ is the thermal permeability coefficient, $m^2$/(s·K).

For the same formation, assuming that the thermal conductivity coefficient $k_T$ is constant [34], and there is no internal heat source in the reservoir, then:

$$\frac{\partial T}{\partial t} = \frac{k_T}{\rho_0 c_p} \cdot \nabla^2 T \tag{13}$$

In this paper, only the reservoir temperature at the distance $r$ from the borehole is required. Therefore, Equation (13) degenerates into a one-dimensional heat conduction equation, that is, a parabolic equation. The analytical solution form can be obtained by

$$T = T_{\mathrm{w}} - (T_{\mathrm{w}} - T_0)erf\frac{r}{2\sqrt{K_{\mathrm{tdf}}t}} \tag{14}$$

Substitute Equation (12) into continuity equation, then:

$$\phi\frac{\partial \rho_{\mathrm{f}}}{\partial t} - \frac{k\rho_{\mathrm{f}}}{\mu}\nabla^2 p + \frac{\rho_{\mathrm{f}}\rho_0 c_{\mathrm{p}}K_{\mathrm{T}}}{k_{\mathrm{T}}}\frac{\partial T}{\partial t} - \frac{k}{\mu}\nabla p\nabla \rho_{\mathrm{f}} + K_T\nabla T\nabla \rho_{\mathrm{f}} = 0 \tag{15}$$

where $\frac{k\rho_{\mathrm{f}}}{\mu}\nabla^2 p$, $\frac{k}{\mu}\nabla p\nabla \rho_{\mathrm{f}}$ and $k_{\mathrm{T}}\nabla T\nabla \rho_{\mathrm{f}}$ are small quantities which can be ignored, and set the coupling coefficient $c_{\mathrm{cou}} = \frac{\rho_0 c_{\mathrm{p}}K_{\mathrm{T}}}{k_{\mathrm{T}}}$, then:

$$\phi\frac{\partial \rho_{\mathrm{f}}}{\partial t} + c\rho_{\mathrm{f}}\frac{\partial T}{\partial t} = 0 \tag{16}$$

Equation (13), Equation (16), and $CO_2$ physical property equation are comprehensively solved to obtain the change relationship of reservoir temperature and pore pressure with time and location.

The boundary conditions are

$$\begin{cases} p(r = R_{\mathrm{w}}, t > 0) = p_{\mathrm{w}} \\ T(r = R_{\mathrm{w}}, t > 0) = T_{\mathrm{w}} \\ p(r \to \infty, t > 0) = p_{\mathrm{po}} \\ T(r \to \infty, t > 0) = T_0 \\ p(r, t = 0) = p_0 \\ T(r, t = 0) = T_0 \end{cases} \tag{17}$$

## 3. Model Solving and Verification

### 3.1. Model Solving Method

Due to the compressibility of $CO_2$, its physical parameters are sensitive to the changes in temperature and pressure. At the same time, the physical parameters of $CO_2$ also change the distribution of reservoir temperature and pore pressure near the well, thus affecting the local stress state of the reservoir. $CO_2$ interacts with rock after invasion into reservoir, and affects the physical and mechanical properties of rock [35,36]. Therefore, when calculating the fracture initiation pressure, it is necessary to comprehensively consider the following parameters such as influence of $CO_2$ on the physical and mechanical properties of rocks, fluid seepage and heat transfer, $CO_2$ physical property changes, and other factors before conducting a coupling solution. The calculation process is shown in Figure 1. In this paper, the Span Wagner equation [37] and Vesovic equation [38,39] with internationally recognized high accuracy are used to calculate the thermophysical properties (density, specific heat capacity, etc.) and migration properties (viscosity, thermal conductivity, etc.) of $CO_2$ to ensure the accuracy of the model.

### 3.2. Model Validation

This paper uses the SC-$CO_2$ fracturing field data from a tight sandstone gas well to verify the model (Table 1). The field data show that the fracturing occurred at the initial stage of injection, the initial pressure decreases from 66.02 MPa to 48.62 MPa, the bottom hole temperature was 368.97 K, and the original formation temperature was 382.15 K. The fracture initiation pressure calculated by the model in this paper is 48.94 MPa, and the absolute error is about 0.32 MPa, which is within 1%, indicating that the model in this paper has high calculation accuracy.

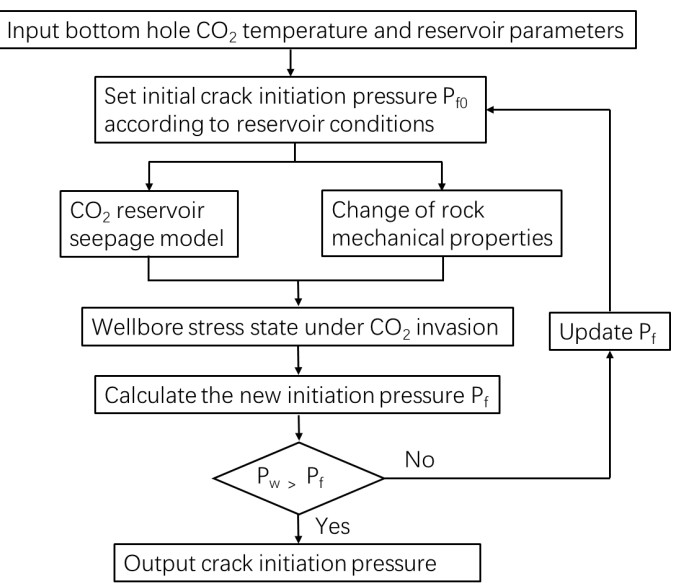

**Figure 1.** Coupling calculation process.

**Table 1.** Basic parameters.

| Parameter | Value |
|---|---|
| Well depth/m | 3 250 |
| Borehole diameter/mm | 121.36 |
| Formation pressure/MPa | 28.16 |
| Inlet temperature/K | 258 |
| Wellhead pressure/MPa | 8 |
| Ground temperature gradient/$(K \cdot 100\ m^{-1})$ | 2.8 |
| Formation thermal expansion coefficient/$K^{-1}$ | $3 \times 10^{-5}$ |
| Thermal permeability coefficient/$(m^2 \cdot (s \cdot K)^{-1})$ | $6.0 \times 10^{-11}$ |
| Original fracture initiation pressure/MPa | 66.02 |
| Formation tensile strength/MPa | 11.256 |
| Elastic modulus/GPa | 22.04 |
| Poisson's ratio | 0.312 |
| Porosity | 0.05 |
| Maximum horizontal stress gradient/$(g \cdot cm^{-3})$ | 1.9 |
| Minimum horizontal stress gradient/$(g \cdot cm^{-3})$ | 1.5 |
| Vertical stress gradient/$(g \cdot cm^{-3})$ | 2.3 |
| Coupling coefficient | $1.2375 \times 10^{-3}$ |

## 4. Results

### 4.1. Distribution of Formation Temperature and Pressure

During the fracturing and injection of SC-$CO_2$, the bottom hole temperature gradually decreases and is lower than the formation temperature, while the bottom hole pressure is higher than the pore pressure, so $CO_2$ is easily invades the formation, affecting the formation temperature and pore pressure near the well [40,41]. Previous research shows that the fracturing fluid injection rate is the main control factor affecting the temperature and pressure of $CO_2$ in the wellbore [40]. The bottom hole temperature decreases with the increase in injection rate. This paper refers to the calculation of bottom hole temperature and pressure in reference [29] and analyzes the distribution rule of formation temperature and pore pressure near the well under different injection rate (0.5 m$^3$/min, 2.0 m$^3$/min) (Figures 2 and 3).

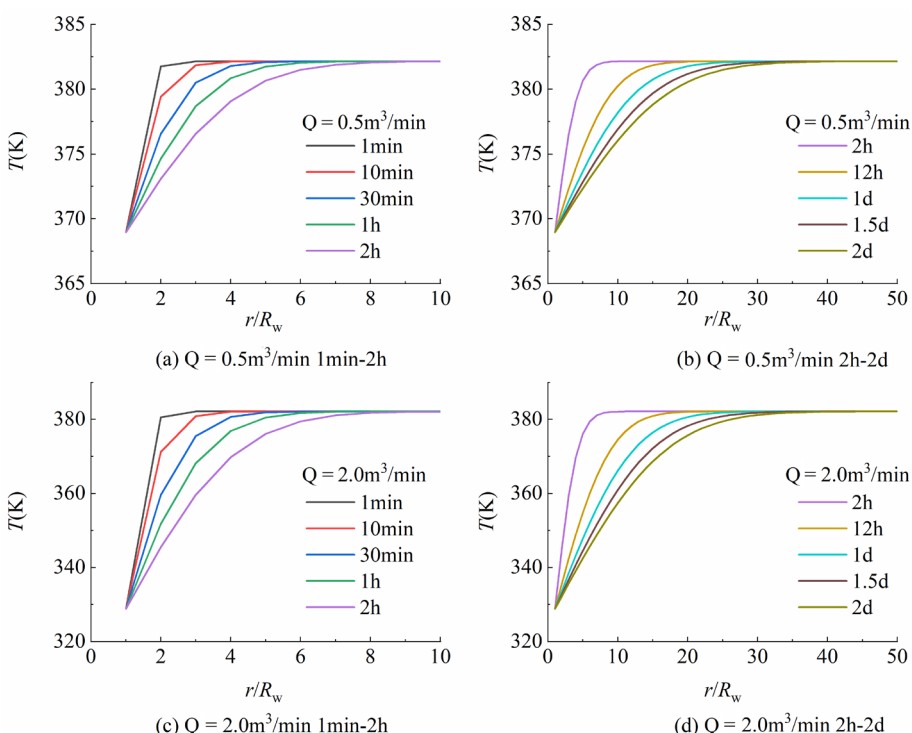

**Figure 2.** Distribution of formation temperature near well under different injection rate.

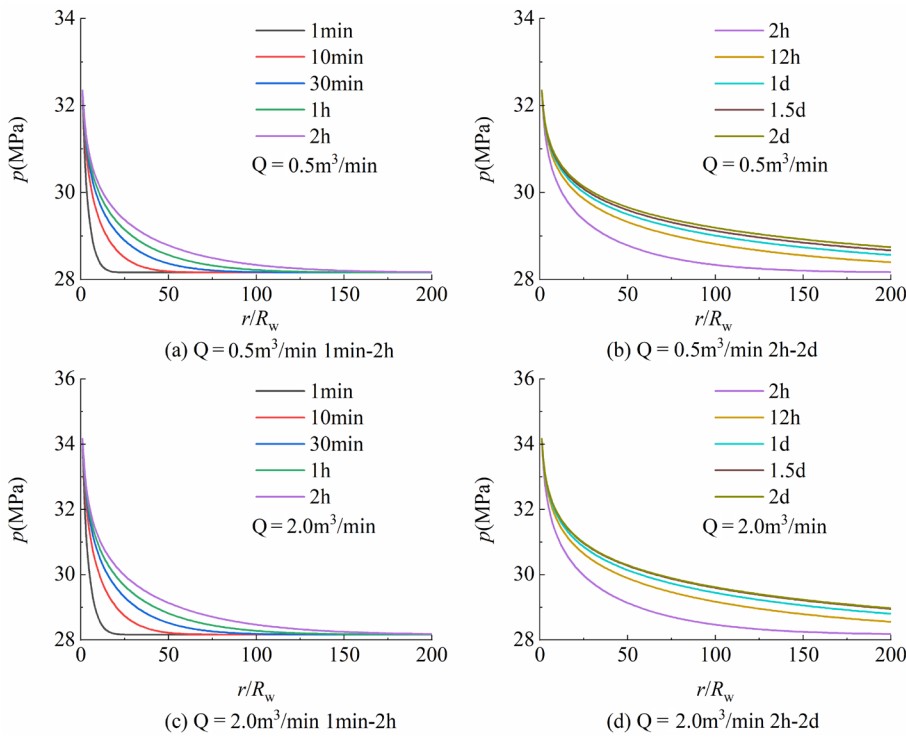

**Figure 3.** Distribution of pore pressure near well under different injection rate.

It can be seen from Figure 2 that when the injection rate increases from 0.5 m³/min to 2.0 m³/min, the bottom hole temperature decreases from 368.97 K to 328.83 K, a decrease of 40.14 K. Under a different injection rate, the formation near the bottom of the well has an evident temperature drop with the increase in time, and the influence range gradually expands. In addition, it can be seen from Figure 2 that the injection rate has little influence on the range of formation temperature drop.

Figure 3 shows the distribution of pore pressure near the wellbore formation with time and radial distance under different injection rate (0.5 m³/min, 2.0 m³/min). It can be seen from Figure 3 that under the same wellhead pressure, when the injection rate increases from 0.5 m³/min to 2.0 m³/min, the bottom hole pressure increases from 32.35 MPa to 34.17 MPa, an increase of 1.82 MPa. This is because the pressure increment caused by the increase in gravity is greater than the frictional pressure drop increment caused by the increase in injection rate. Under a different injection rate, the formation pore pressure near the bottom hole increases with the increase in time, and the influence range increases. In addition, it can be found that the sweep velocity of bottom hole pressure is significantly higher than that of bottom hole temperature by comparing Figures 2 and 3.

### 4.2. Distribution of In Situ Stress

The near-wellbore temperature and pore pressure are substituted into Equations (5) and (7) to obtain the stress changes caused by formation temperature and pore pressure (Figure 4). It can be seen from Figure 4a that the thermal stress near the well wall ($\sigma_{\theta 4}$) is negative, which can effectively reduce the tangential stress at the well wall, thus reducing the fracture initiation pressure. Maximum temperature stress ($|\sigma_{\theta 4}|$) is 5.24 MPa, which is located at the well wall. The absolute value of temperature stress gradually decreases and changes to a positive value with the increase in radial distance; the negative range of temperature stress increases with the increase in time.

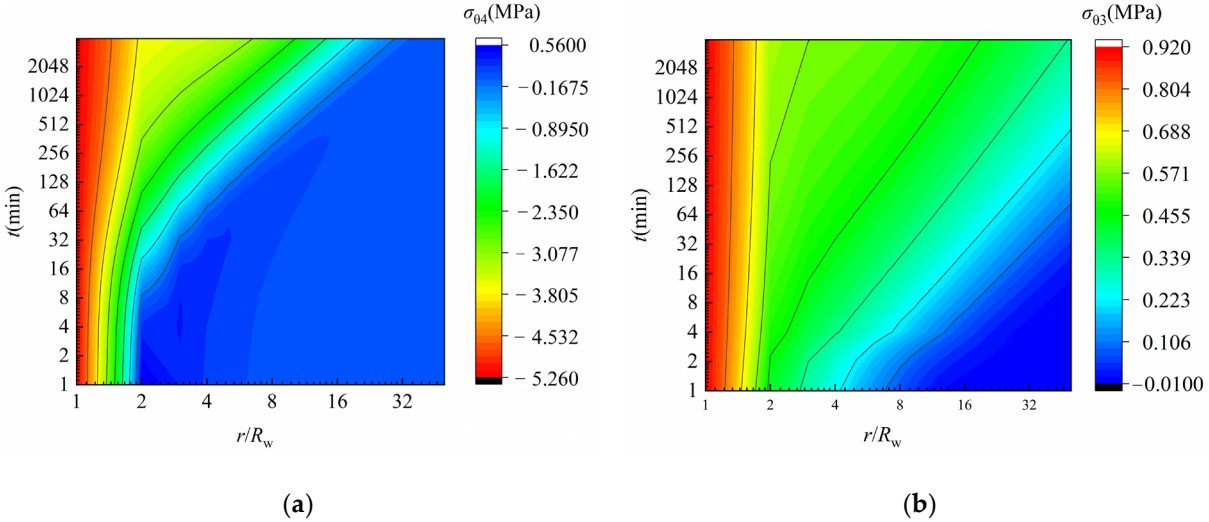

**Figure 4.** Rock stress changes caused by formation temperature and pore pressure (Q = 0.5 m³/min). (**a**) Formation temperature; (**b**) pore pressure.

It can be seen from Figure 4b that the tangential stress caused by pore pressure near the wellbore ($\sigma_{\theta 3}$) is a positive value, indicating that the increase in pore pressure can cause the increase in tangential stress and the increase in fracture initiation pressure. The maximum value of $\sigma_{\theta 3}$ also appears at the well wall, which is 0.92 MPa. In the process of $CO_2$ fracturing, the effect of pore pressure change on the tangential stress at the well wall is less than that of the reservoir temperature; as the radial distance increases, $\sigma_{\theta 3}$ gradually decreases and turns into a negative value. The range of positive values of $\sigma_{\theta 3}$ increases gradually with the increase in time. In addition, it can be found from Figure 4b that the tangential stress concentration area caused by pore pressure is concentrated near the well wall.

In general, the stress caused by the change in formation temperature during SC-$CO_2$ fracturing is greater than that caused by the change in pore pressure. This is because the injected low temperature $CO_2$ can reach the supercritical state when reaching the bottom of

the well, but the temperature is lower than the reservoir temperature, resulting in greater thermal stress.

Figure 5 shows the distribution of tangential stress in the near-wellbore formation. It can be seen from the figure that the minimum tangential stress is 41.16 MPa, which is located at the well wall. With the increase in the radial distance, the tangential stress first increases, then decreases and gradually plateaus. There is a stress concentration area within the range of 2~4 times the borehole diameter. With the increase in time, the stress concentration area is gradually released, which is the result of the comprehensive effect of formation temperature and pore pressure changes near the well.

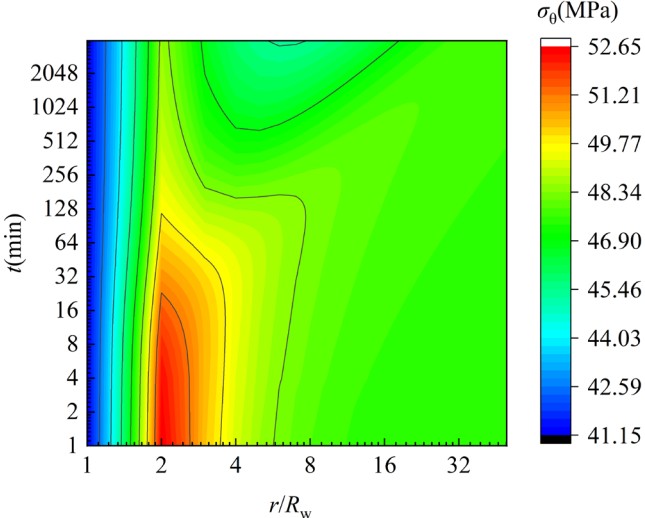

**Figure 5.** Distribution of formation tangential stress (Q = 0.5 $m^3$/min).

### 4.3. Influencing Factors of Fracture Initiation Pressure

It can be seen from Equation (10) that there are many factors that affect the fracture initiation pressure of SC-$CO_2$ fracturing. In this paper, the effects of rock mechanical properties (Poisson's ratio, elastic modulus) and bottom hole temperature are analyzed.

#### 4.3.1. Poisson's Ratio

After the invasion of $CO_2$, the axial strain of reservoir rock increases, leading to a decrease in Poisson's ratio. Ding et al. [30] found through shale core experiments that the Poisson's ratio of reservoir rock decreases from 0.280 to 0.257, a decrease of 8.2 % after the invasion of $CO_2$.

Porous elastic stress coefficient $\eta$ and thermal stress $\sigma_{\theta T}$ are affected by Poisson's ratio. It can be seen from Equations (5) and (7) that the former decreases, while the latter increases with the increase in Poisson's ratio. Figure 6 shows the variation in the porous elastic stress coefficient and the fracture initiation pressure with Poisson's ratio under different Biot coefficients.

It can be seen from Figure 6 that $\eta$ and pf increase with the increase in Biot coefficient $\alpha$. Under different Biot coefficients, $\eta$ and pf reduce with the increase in Poisson's ratio of rock. Combined with the research results of Ding et al. [34], the Poisson's ratio of rocks after $CO_2$ invasion is reduced, so the variation in $CO_2$ into the formation may increase the initial fracturing pressure of the reservoir.

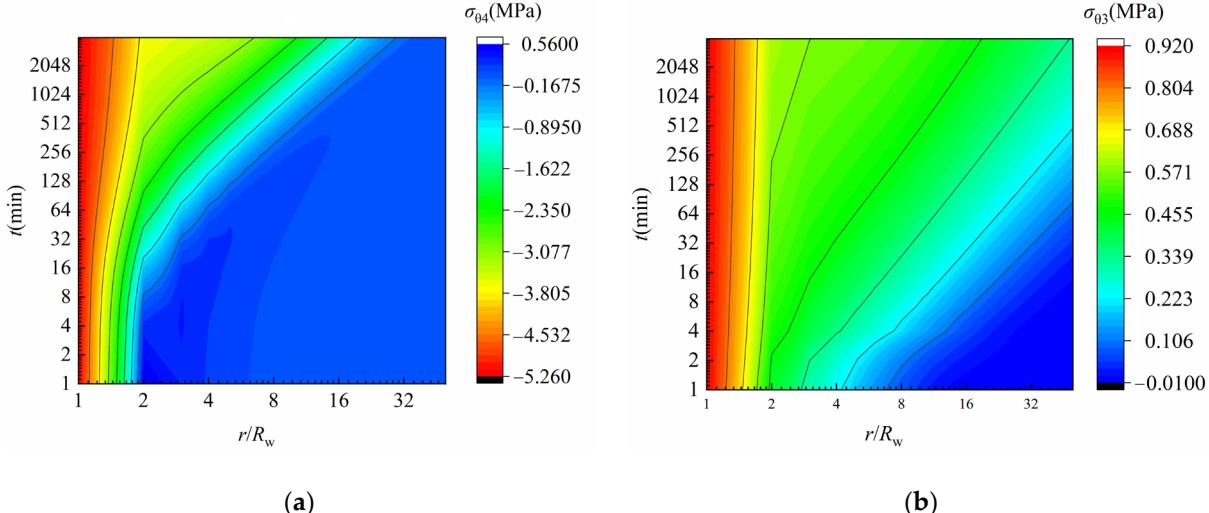

**Figure 6.** Change in elastic stress coefficient and fracture initiation pressure with Poisson's ratio. (**a**) Elastic stress coefficient; (**b**) fracture initiation pressure.

### 4.3.2. Elastic Modulus

Ding et al. found that the strain difference that the core can withstand decreases and the elastic modulus increases after intrusion of $CO_2$. According to Equation (10), when the bottom hole fluid temperature is low, the initiation pressure decreases with the increase in elastic modulus. Figure 7 shows the change in the fracture initiation pressure with the elastic modulus. It can be seen from the figure that the fracture initiation pressure decreases linearly with the increase in elastic modulus.

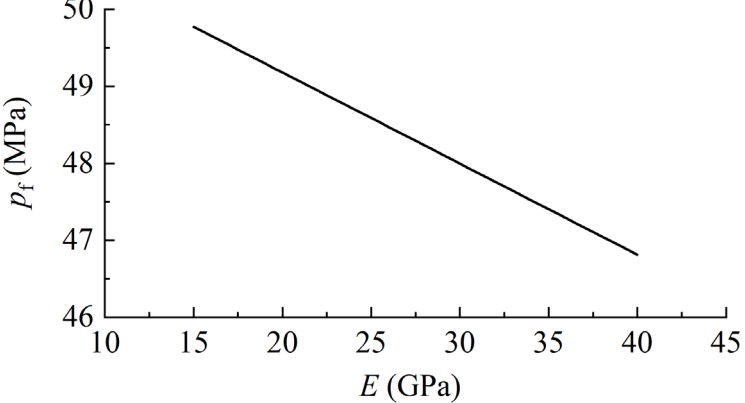

**Figure 7.** Change in initiation pressure with elastic modulus.

### 4.3.3. Bottom Hole Temperature

Bottom hole temperature is an important factor affecting the fracture initiation pressure of SC-$CO_2$ fracturing reservoirs. The change in initiation pressure with Poisson's ratio at different bottom hole temperatures is shown in Figure 8. In the figure, the injection rate is 0.5 m$^3$/min and 2.0 m$^3$/min, respectively, and the corresponding bottom hole temperature difference and formation temperature difference are 13.18 K (bottom hole temperature 368.97 K) and 53.32 K (bottom hole temperature 328.83 K), respectively.

It can be seen from Figure 8 that under different temperature differences, the reservoir fracture initiation pressure decreases linearly with the increase in Poisson's ratio. The reservoir pressure decreases with the increase in temperature difference between the bottom hole and the reservoir.

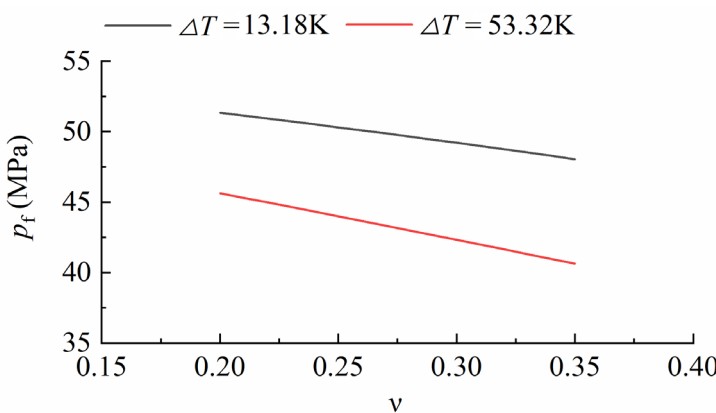

**Figure 8.** Effect of temperature difference between fluid and formation on initiation pressure.

According to the analysis above, the Poisson's ratio and elastic modulus of reservoir rock change distinctly after contacting with $CO_2$. In order to comprehensively analyze the influence of changes in Poisson's ratio and elastic modulus of rock on the initiation pressure, the contour of the initiation pressure changes with Poisson's ratio and elastic modulus when the bottom hole temperature is 368.97 K as provided in Figure 9. It can be seen from the figure that the increase in Poisson's ratio and elastic modulus can lead to the decrease in reservoir initiation pressure. In practical application, the change in initiation pressure can be judged according to the increment of Poisson's ratio and elastic modulus after $CO_2$ invasion.

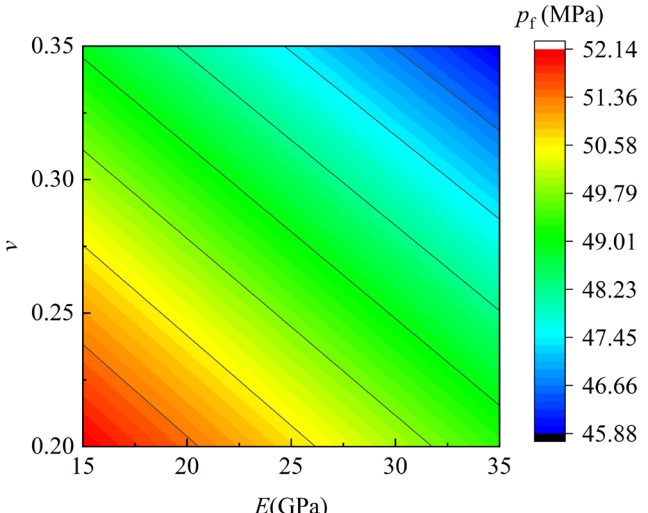

**Figure 9.** Effect of Poisson's ratio and elastic modulus on initiation pressure.

## 5. Application Example

Based on the analysis above, the initiation pressure at the injection rate of 0.5 m$^3$/min and 2.0 m$^3$/min is calculated, respectively (Table 2). It can be seen from Table 2 that SC- $CO_2$ fracturing can effectively reduce the initial pressure of the reservoir. When the injection rate increases from 0.5 m$^3$/min to 2.0 m$^3$/min, the initial pressure decreases from 48.94 MPa to 40.99 MPa, reducing by nearly 8 MPa. It should be noted that although increasing the injection rate can reduce the fracture initiation pressure, it also brings large energy consumption. In the fracture initiation stage, it is not appropriate to excessively pursue the low fracture initiation pressure under high injection rate.

**Table 2.** Calculation results of initiation pressure at the injection rate of 0.5 m$^3$/min and 2.0 m$^3$/min.

| Q /(m$^3$·min$^{-1}$) | Bottom Hole Temperature /K | Initiation Pressure /MPa | Original Initiation Pressure /MPa | Reduction Rate /% |
|---|---|---|---|---|
| 0.5 | 368.97 | 48.94 | 66.02 | 25.87 |
| 2.0 | 328.83 | 40.99 | 66.02 | 37.91 |

From the perspective of dynamic fracturing process, low initial pressure is conducive to the formation of fractures, which expands rapidly under the bottom hole pressure. Due to the slow propagation speed of the temperature field, the thermal stress generated by the temperature field can induce the generation and expansion of secondary fractures, and thus create a fracture network system with high conductivity in the reservoir.

### 6. Conclusions

SC-CO$_2$ fracturing can effectively reduce the fracture initiation pressure of reservoir. The fracture initiation pressure decreases with the increase in injection rate.

With the injection of CO$_2$, the influence of bottom hole temperature and pressure on the near-wellbore formation expands radially. The sweep velocity of bottom hole pressure is higher than that of bottom hole temperature. The injection rate mainly affects the near-wellbore formation temperature and pore pressure, but has little influence on the sweep velocity.

The maximum tangential stress caused by formation temperature and pore pressure are both located at the well wall. The former reduces the tangential stress at the well wall, while the latter increases the tangential stress at the well wall. The stress caused by formation temperature change is greater than that caused by pore pressure change. On the whole, the minimum tangential stress is also located at the well wall.

The increase in Poisson's ratio, the increase in elastic modulus and the decrease in bottom hole temperature can all contribute to reducing the initial pressure of the reservoir.

**Author Contributions:** Conceptualization and methodology, Y.D.; validation, formal analysis, investigation, resources, data curation and writing—original draft preparation, Y.C., Z.K., Y.K., X.C. (Xiaocheng Chen), X.C. (Xiaohong Chen), Q.F. and J.W.; writing—review and editing, visualization, supervision and project administration, Y.D. All authors have read and agreed to the published version of the manuscript.

**Funding:** This research was supported by the National Natural Science Foundation of China (51404287) and Fundamental Research Funds for the Central Universities (18CX02072A).

**Data Availability Statement:** Not applicable.

**Conflicts of Interest:** The authors declare no conflict of interest.

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
