# Peer review of "A Thermal-Fluid-Solid Coupling Computation Model of Initiation Pressure Using Supercritical Carbon Dioxide Fracturing"

_processes, doi:10.3390/pr11020437_

Round 1

Reviewer 1 Report

The paper “A Thermal-fluid-solid Coupling Computation Model of Initiation Pressure using Supercritical Carbon Dioxide Fracturing” presents a mathematical model to calculate fracture initiation pressure. The model is relatively simple and accounts for the stress variation due to far field stresses, as well as poro-thermoelastic changes caused by CO2. Results demonstrate that the aforementioned thermo-poroelastic effects are able to reduce the initiation pressure.

I think that this is an interesting paper, but definitely not groundbreaking. Also, the fracture initiation also depends on perforation orientation. A brief discussion on how te presence of perforation tunnel affects initiation is necessary.

Author Response

Thank you very much for your careful comment. We quite agree with you that the fracture initiation depends on perforation orientation, which is also our follow-up research of this paper. However, the fracture initiation mechanism considering the effect of perforation orientation is complex, and we plan to discuss in the subsequent work. Thank you again for your professional advice.

Reviewer 2 Report

(1) What is the implementation platform used this study? COMSOL? ABAQUS? Or Matlab? If it is first two commercial platforms mentioned above, how much can the basic theory contribute to the research?

(2) We know that the basic theoretical research of hydraulic fracturing will only include two parts: seepage mechanics and solid mechanics. The temperature field part is usually not considered. The phase transition of supercritical carbon dioxide is affected by temperature. It is right to consider the influence of temperature field when studying supercritical carbon dioxide fracturing. However, in the model verification, it is better to give the fracturing effect with and without temperature effect by comparison.

(3) This study is a numerical model of coupling of three physical fields, and its nonlinearity must be very strong. However, in Figure 7, the initiation pressure is a nearly linear relationship of elastic modulus. The reviewer thinks it is unreasonable. Please recalculate or explain.

(4) In section 4, there is only a description of the results, but no explanation of the mechanism. The reviewer suggested that the mechanism of each part be explained. If possible, give a schematic diagram of the mechanism.

(5) It is worth adding references to support your statement “Unconventional oil and gas resources in China are characterized by tight reservoir, complex geological structure and high clay mineral content, which increases the difficulty of unconventional oil and gas field development.” The following referencs can be refered to and cited. https://doi.org/10.3389/fenrg.2022.919966, https://doi.org/10.1007/s40948-022-00396-0, https://doi.org/10.1007/s11356-021-18169-9, 

(6) There are still some negligible problems in the manuscript that need to be corrected. â‘ When conducting research related to hydraulic fracturing, we usually use fracture instead of crack. â‘¡  The language needs to be properly polished by native English speakers. â‘¢ The volume of fracturing fluid used in fracturing fluid is usually the injection rate rather than the displacement.

Author Response

Responses to Reviewer #2 Comments

----------------------------------------------------------------------------------------------------------------------

  1. What is the implementation platform used this study? COMSOL? ABAQUS? Or Matlab? If it is first two commercial platforms mentioned above, how much can the basic theory contribute to the research?

Response: Thank you very much for your careful comment. We used Matlab for the calculation in this study, and the basic parameters have been displayed in the paper.

  1. We know that the basic theoretical research of hydraulic fracturing will only include two parts: seepage mechanics and solid mechanics. The temperature field part is usually not considered. The phase transition of supercritical carbon dioxide is affected by temperature. It is right to consider the influence of temperature field when studying supercritical carbon dioxide fracturing. However, in the model verification, it is better to give the fracturing effect with and without temperature effect by comparison.

Response: Thank you very much for your careful comment. We have compared the fracturing effect with and without temperature in Section 3.2 in the revised manuscript.

  1. This study is a numerical model of coupling of three physical fields, and its nonlinearity must be very strong. However, in Figure 7, the initiation pressure is a nearly linear relationship of elastic modulus. The reviewer thinks it is unreasonable. Please recalculate or explain.

Response: Thank you very much for your professional comment. In order to find out the influence of different factors on initiation pressure, we use control variate method during the calculation. And in Figure 7, the variate is Elastic Modulus, all other parameters are fixed. We also notice that the initiation pressure is a nearly linear relationship of elastic modulus in our work. According to Equation 9 in the manuscript, we can get the relationship between the initiation pressure and the elastic modulus, as shown in Eq. (1). We can conclude that the elastic modulus is linearly related to the initiation pressure when other parameters are fixed, and the elastic modulus affects only the thermal stress.

                                                          (1)

  1. In section 4, there is only a description of the results, but no explanation of the mechanism. The reviewer suggested that the mechanism of each part be explained. If possible, give a schematic diagram of the mechanism.

Response: Thank you very much for your careful comment. In section 4, we described the distribution of formation temperature, pressure and in-situ stress, influencing factors of fracture initiation pressure. The fracture initiation model for carbon dioxide fracturing considering the bottom hole pressure and temperature condition was discussed in detail in another paper in which the mechanism was explained. Thank you again for your professional advice.

  1. It is worth adding references to support your statement “Unconventional oil and gas resources in China are characterized by tight reservoir, complex geological structure and high clay mineral content, which increases the difficulty of unconventional oil and gas field development.” The following referencs can be refered to and cited. https://doi.org/10.3389/fenrg.2022.919966, https://doi.org/10.1007/s40948-022-00396-0, https://doi.org/10.1007/s11356-021-18169-9, 

Response: Thank the reviewer for your careful comments. We have added relevant references in the introduction.

  1. There are still some negligible problems in the manuscript that need to be corrected. â‘ When conducting research related to hydraulic fracturing, we usually use fracture instead of crack. â‘¡ The language needs to be properly polished by native English speakers. â‘¢ The volume of fracturing fluid used in fracturing fluid is usually the injection rate rather than the displacement.

Response: We are really grateful for the reviewer’s professional advice. We have corrected the negligible problems in the manuscript.

Reviewer 3 Report

By establishing a "thermal-fluid-solid" coupling fracturing model, the authors studied the changes of formation temperature and pore pressure near the wellbore caused by CO2 invasion into the formation, as well as its influence on rock mechanical parameters. The structure and logic of the paper meet the requirements of the journal. It is suggested to make the following revise before publication:

1. It is mentioned in the last paragraph of article "1 Introduction" that SC-CO2 is currently limited to laboratory experiments and that there are certain deficiencies in the research on theoretical models. The author should strengthen the summary describing the deficiencies of others' research and analyze the highlights of his own research.

2. In "4.1. Distribution of Formation Temperature and Pressure", is the flow rate of 0.5 and 2.0 m3/min consistent with the flow rate in oilfield fracturing? In addition, can you quantify how many meters near the wellbore the formation temperature is more affected?

3. In "5. Application Example", the author describes the influence of temperature and flow rate on crack initiation pressure, but a low crack initiation pressure does not necessarily form a complex fracture network structure. The author should add to the description of the complexity of the cracks.

4. The author should list the shortcomings of this study and the direction of further research in the summary.

5. It is suggested that the author further modify the grammar of the full text. Most of the references are not from the last five years, so authors are advised to refer to the latest references related to their own research.

Author Response

  1. It is mentioned in the last paragraph of article "1 Introduction" that SC-CO2 is currently limited to laboratory experiments and that there are certain deficiencies in the research on theoretical models. The author should strengthen the summary describing the deficiencies of others' research and analyze the highlights of his own research.

Response: Thank you very much for your careful comment. The highlights of our research is that we want to establish a thermal-fluid-solid coupling SC-CO2 fracturing initiation pressure model to calculate the initiation pressure and provide theoretical guidance for on-site construction. The SC-CO2 fracture initiation mechanism considering the effect of multiple factors is complex, and we plan to further discuss in the subsequent work. Thank you again for your professional advice.

  1. In "4.1. Distribution of Formation Temperature and Pressure", is the flow rate of 0.5 and 2.0 m3/min consistent with the flow rate in oilfield fracturing? In addition, can you quantify how many meters near the wellbore the formation temperature is more affected?

Response: We are really grateful for the reviewer’s professional advice. First, because of the strong compressibility of carbon dioxide, the injecting process can be divided into two stages, which are pre-initiation (before the fracture initiating) and post-initiation (after the fracture initiating). While pre-initiation is considered in this paper. According to the field data and former researches [1, 2], the pump rate during the pre-initiation is lower and generally not larger than 2.0 m3/min. Second, this is an interesting question. We will improve the calculation procedure in the following work and try to quantify the distance of the formation temperature affected. Thank you again for your helpful advice.

-----------------------------------

[1] Xiao C., Ni H., Shi X. Unsteady model for wellbore pressure transmission of carbon dioxide fracturing considering limited-flow outlet. Energy 2021, 122289.

[2] Xiao C., Ni H., Shi X., et al. A fracture initiation model for carbon dioxide fracturing considering the bottom hole pressure and temperature condition. Journal of Petroleum Science and Engineering 2020, 184, 106541.

  1. In "5. Application Example", the author describes the influence of temperature and flow rate on crack initiation pressure, but a low crack initiation pressure does not necessarily form a complex fracture network structure. The author should add to the description of the complexity of the cracks.

Response: Thank you for your careful advice. In "5. Application Example", we conclude that increasing the pump rate can effectively reduce the initiation pressure for supercritical carbon dioxide fracturing, while excessive pump rate might bring large energy consumption, which is not necessary during pre-initiation stage for supercritical carbon dioxide fracturing. In addition, the complexity of the fractures is not involved in this paper.

  1. The author should list the shortcomings of this study and the direction of further research in the summary.

Response: Thank you for your professional advice. We didn’t couple the perforation tunnel and the deviation/azimuth angle in the present model, and we will consider the above factors in the theoretical model in our further researches.

  1. It is suggested that the author further modify the grammar of the full text. Most of the references are not from the last five years, so authors are advised to refer to the latest references related to their own research.

Response: Thank the reviewer for your careful comments. We have modified the grammar and added relevant latest references in the introduction.

Round 2

Reviewer 2 Report

The quality of the manuscript has been greatly improved, and it can be considered to be accepted and published.